# The Effect of Low Doses of Zearalenone (ZEN) on the Bone Marrow Microenvironment and Haematological Parameters of Blood Plasma in Pre-Pubertal Gilts

**DOI:** 10.3390/toxins14020105

**Published:** 2022-01-29

**Authors:** Magdalena Mróz, Magdalena Gajęcka, Katarzyna E. Przybyłowicz, Tomasz Sawicki, Sylwia Lisieska-Żołnierczyk, Łukasz Zielonka, Maciej Tadeusz Gajęcki

**Affiliations:** 1Department of Veterinary Prevention and Feed Hygiene, Faculty of Veterinary Medicine, University of Warmia and Mazury in Olsztyn, Oczapowskiego 13/29, 10-718 Olsztyn, Poland or magdalena.mroz@uwm.edu.pl (M.M.); lukaszz@uwm.edu.pl (Ł.Z.); gajecki@uwm.edu.pl (M.T.G.); 2Department of Human Nutrition, Faculty of Food Sciences, University of Warmia and Mazury in Olsztyn, Słoneczna 45F, 10-719 Olsztyn, Poland; katarzyna.przybylowicz@uwm.edu.pl (K.E.P.); tomasz.sawicki@uwm.edu.pl (T.S.); 3Independent Public Health Care Centre of the Ministry of the Interior and Administration, and the Warmia and Mazury Oncology Centre in Olsztyn, Wojska Polskiego 37, 10-228 Olsztyn, Poland; lisieska@wp.pl

**Keywords:** zearalenone, low dose, bone marrow microenvironment, haematology, pre-pubertal gilts

## Abstract

The aim of this study was to determine whether low doses of zearalenone (ZEN) influence the carry-over of ZEN and its metabolites to the bone marrow microenvironment and, consequently, haematological parameters. Pre-pubertal gilts (with a body weight of up to 14.5 kg) were exposed to daily ZEN doses of 5 μg/kg BW (group ZEN5, *n* = 15), 10 μg/kg BW (group ZEN10, *n* = 15), 15 μg/kg BW (group ZEN15, *n* = 15), or were administered a placebo (group C, *n* = 15) throughout the entire experiment. Bone marrow was sampled on three dates (exposure dates 7, 21, and 42—after slaughter) and blood for haematological analyses was sampled on 10 dates. Significant differences in the analysed haematological parameters (WBC White Blood Cells, MONO—Monocytes, NEUT—Neutrophils, LYMPH—Lymphocytes, LUC—Large Unstained Cells, RBC—Red Blood Cells, HGB—Haemoglobin, HCT—Haematocrit, MCH—Mean Corpuscular Volume, MCHC—Mean Corpuscular Haemoglobin Concentrations, PLT—Platelet Count and MPV—Mean Platelet Volume) were observed between groups. The results of the experiment suggest that exposure to low ZEN doses triggered compensatory and adaptive mechanisms, stimulated the local immune system, promoted eryptosis, intensified mycotoxin biotransformation processes in the liver, and produced negative correlations between mycotoxin concentrations and selected haematological parameters.

## 1. Introduction

Zearalenone (ZEN) and its metabolites (ZELs), α-zearalenol (α-ZEL) and β-zearalenol (β-ZEL), are the most ubiquitous mycotoxins in plant materials. They are frequently referred to as xenobiotics. These mycotoxins disrupt reproductive functions because they are structurally similar to oestradiol [1,2]. Alpha-ZEL is the main ZEN metabolite that affects pigs. Other animal species, including broilers, cows, and sheep, are more susceptible to β-ZEL, which is characterized by lower levels of metabolic activity [3]. Zearalenone’s activity is determined by biotransformation processes in plants and animals as well as the immune status of the reproductive system (due to variations in steroid hormone levels during maturation, reproduction, and pregnancy) and the gastrointestinal system of the exposed individuals [4,5,6,7].

Different doses of mycotoxins induce various effects [7]. The symptoms and toxic effects of high mycotoxin doses have been extensively researched [8]. Low monotonic doses of mycotoxins are well tolerated by animals during prolonged exposure, and these compounds can meet the animals’ life needs or exert therapeutic effects [9,10,11]. The adverse effects of low mycotoxin doses have also been well documented [12]. These effects can be attributed to hormesis, namely exposure to the highest doses below the No Observed Adverse Effect Level (NOAEL) that induce subclinical (asymptomatic) states or interact positively with the host organism in various stages of life [13,14,15,16]. In turn, doses corresponding to the Lowest Observed Adverse Effect Level (LOAEL) induce clear clinical symptoms of mycotoxicosis. The lowest measurable dose that interacts positively with the host body in different stages of life is known as the Minimal Anticipated Biological Effect Level (MABEL) [16].

The dose-response relationship has been also undermined by the low dose hypothesis. The above applies particularly to hormonally active chemical compounds, including mycoestrogens such as ZEN and ZELs, which disrupt the functioning of the endocrine system (ED), even when ingested in small quantities [5,17]. This ambiguous dose-response relationship does not justify direct analyses or meta-analyses of the risk (clinical symptoms or the results of laboratory analyses) associated with the transition from high to low doses [18]. The concept of the lowest identifiable dose that produces counter-intuitive effects is becoming increasingly popular in biomedical sciences. For this reason, the relevant mechanisms should be investigated to support rational decision-making in selected processes [7,19]. Substances that both disrupt and contribute to homeostasis have undermined the traditional concepts in toxicology, in particular “the dose makes the poison” adage. Zearalenone and ZELs induce different responses in mammals when administered in low doses [13]. Similar observations have been made by Knutsen et al. [12]. According to the Scientific Panel on Contaminants in the Food Chain (CONTAM), the influence of ZEN on animal health should be re-evaluated based on the animals’ responses to the lowest detectable doses of ZEN (MABEL; the highest values of NOAEL and LOAEL), including the parent compound and its metabolites, in feed [7,12,20,21].

Hematopoietic stem cells (controlled primarily by the local bone marrow microenvironment as well as by long-range signals from, for example, the endocrine system [22], such as oestrogens or xenobiotics) cannot cross the bone marrow microenvironment barrier. Only mature blood cells containing specific membrane proteins are able to bind to the vascular endothelium and reach peripheral blood. Blood vessels act as a barrier that prevents immature blood cells from leaving the bone marrow microenvironment. Different types of haematopoietic cells have specific roles: red blood cells (RBC) contain haemoglobin, a specific protein that transports oxygen within the body [23], white blood cells (WBC) participate in immune functions and help fight infections, and platelets (PLT) participate in blood clotting at the sites of vascular injury [24].

The bone marrow microenvironment is an interesting object of scientific inquiry [25]. Red marrow is highly vascular, soft, gelatinous-spongy tissue, which fills the cavities of long bones and the spaces between trabeculae in cancellous bones, such as the wing of ilium. Red marrow is the primary producer of blood cells in the body, and it actively participates in haematopoiesis. Each day, red marrow produces 200 billion new morphotic elements in the blood. It is composed of delicate, highly vascular, fibrous tissue, and it contains hematopoietic stem cells, which give rise to various types of blood cells that are transported to the bloodstream upon maturation [26]. Yu and Scadden [27] described bone marrow as a carrier with sub-compartments that support different haematopoietic activities.

The aim of the present study was to determine whether exposure to low ZEN doses (MABEL [5 μg/kg body weight—BW], the highest values of NOAEL [10 μg/kg BW], and LOAEL [15 μg/kg BW]) administered *per os* to pre-pubertal gilts over a period of 6 weeks influences the concentrations of ZEN and ZELs in the bone marrow microenvironment and induces changes in selected haematological parameters.

## 2. Results

The presented results provide the basis for continuing research on the assessment of the impact of very low doses of ZEN on the macroorganism. There is a specific interaction between the mycotoxin level in the bone marrow microenvironment and the hematological values of the test animals. Preliminary data analysis documents that it is a set of highly individualized results for the parent substance and the absence of essential ZEN metabolites. The presented experimental design confirmed the correctness of the use of the method (chromatography), classified as precision medicine: medicine that allows the assessment of a single organism as well as the entire population. The existing values of the concentration of the undesirable substance in the bone marrow microenvironment contributed to the negative values of the Pearson’s r coefficient for white and red blood cells and platelets.

### 2.1. Experimental Feed

The analysed feed did not contain mycotoxins, or its mycotoxin content was below the sensitivity of the method (VBS). The concentrations of masked mycotoxins were not analysed.

### 2.2. Clinical Observations

The results presented in this paper were acquired during a large-scale experiment where clinical signs of ZEN mycotoxicosis were not observed. However, changes in specific tissues or cells were frequently noted in analyses of selected serum biochemical parameters, heart muscle, coronary artery, genotoxicity of caecal water, selected steroid concentrations, gut microbiota parameters, and the animals’ body weight gains. Samples for laboratory analyses were collected from the same pre-pubertal gilts. The results of these analyses were presented in our previous studies [7,16,28,29].

### 2.3. Concentrations of Zearalenone and Its Metabolites in the Bone Marrow Microenvironment

Zearalenone concentrations in the bone marrow microenvironment did not differ significantly between the experimental groups, but the lowest values were noted on the first two exposure dates (5.64 ng/g on D1; 4.69 ng/g on D2) in group ZEN5 (5 µg ZEN/kg BW) (see Table 1). In turn, significant differences at *p* ≤ 0.05 were observed between group ZEN5 and group ZEN10 (7.74 ng/g and 7.35 ng/g, respectively) on D3. Significant differences at *p* ≤ 0.01 were noted between group ZEN5 and group ZEN15 (7.74 ng/g and 7.03 ng/g, respectively) on D3, and in group ZEN15 between D2 and D3 vs. D1 (6.84 ng/g and 7.03 ng/g vs. 8.17 ng/g, respectively). Similar values had been reported in other tissues, such as the heart muscle [19]. Zearalenone metabolites, α -ZEL and β-ZEL, were not detected (values below the sensitivity of the method).

### 2.4. Carry-OVER Factor (CF)

The carry-over of ZEN from the intestinal lumen to the bone marrow microenvironment collected from the wing of ilium (posterior superior iliac spine), was influenced by the administered dose and time of exposure in each group (see Table 1). The lowest values of the CF were noted in group ZEN5 (in particular on D1 and D3 at 7 × 10^−5^ and 6 × 10^−5^, respectively), and the highest CF values were observed in group ZEN15 (in particular on D3 at 98 × 10^−7^). The values of the CF determined in this study were proportionally higher relative to the values noted in the blood serum [28] and the heart muscle [19] of the same animals.

### 2.5. Results of Haematological Analyses

Only statistically significant differences in all animal groups are presented in the figure drawings. The results of this study and the findings of other authors indicate that even very low concentrations of mycotoxins in feed materials can lead to changes in blood homeostasis in pre-pubertal gilts [10,30,31,32]. The significant differences in the haematological parameters of pre-pubertal gilts exposed to various doses of ZEN (MABEL, highest NOAEL values, and LOAEL) for 6 weeks did not exceed the reference range [32,33]. In view of the above, the results noted in group C were unambiguous (see Appendix A), and they could be used as a reference in a risk assessment analysis, where each result is considered within a range of positive control (increase) or negative control (decrease) values during subchronic exposure to the mycotoxin.

### 2.6. Haematological Analyses

#### 2.6.1. General Analysis

Significant differences were rarely noted, mostly in erythrocyte parameters in the middle (see Figure 1) and at the end of the experiment (see Figure 2), and in the percentages of white blood cells on the first and last date of exposure (see Figure 2). At the beginning of the experiment, the analysed parameters were generally lower in group C, whereas the reverse was noted towards the end of the study. Significant differences were not observed on date 9.

#### 2.6.2. Accompanying Factors

The results of this study could have been influenced by several accompanying factors, including (i) the manner of ZEN transmission to the body, (ii) ZEN dose, (iii) and/or the kinetic effects of mycotoxin bioassimilation [13]. The latter can be subdivided into several sub-processes, beginning from mycotoxin extraction from the feed matrix to its absorption, distribution, and deposition in tissues, and mycotoxin modification [15]. This is a very important consideration, but it was not studied in the described experiment.

The above can have two effects: (i) higher demand for energy needed for biotransformation and constitutive growth of pre-pubertal gilts [7,34], and (ii) a milder response to the applied mycotoxin doses in the peripheral vascular system (*vena cava cranialis*) from which samples for metabolic analyses were collected. As a result, the exposure to low doses of ZEN with low values of the CF to intestinal tissues (towards the end of the experiment) and the liver probably induced minor but significant changes in the values of selected haematological parameters. Changes in WBC (White Blood Cells), EOS (Eosinophils), BASO (Basophils), MONO (Monocytes), RBC—Red Blood Cells; HGB—Haemoglobin, and HCT—Haematocrit, values were observed mainly in the first three weeks of exposure (between sampling dates 1 and 6; see Table 2, Table 3 and Table 4).

Significant differences in PLT (Platelet Count) and PLT clump values were also noted on the same dates (see Table 5). These changes were probably induced by compensatory mechanisms [35].

### 2.7. Correlation Coefficients

Linear correlation coefficients (Pearson’s *r*) were calculated to assess the relationships between ZEN concentrations in the bone marrow microenvironment and the values of blood morphotic components (RBC, WBC, and PLT) on different exposure dates and in different experimental groups (see Table 6). These blood components were selected because erythropoiesis, leukopoiesis, and thrombopoiesis take place during the last processes of hemopoiesis in the bone marrow environment. The strength of the examined correlations can be evaluated based on the values presented in Table 6 [36]. A correlation is negative when *r* < 0. The strength of correlations is evaluated on the following scale: *r* < 0 to −0.2, no negative linear correlation; *r* = −0.21 to −0.39, weak negative correlation; *r* = −0.4 to −0.69, moderate negative correlation; *r* = −0.7 to −0.9, relatively strong negative correlation; *r* > −0.9, very strong negative correlation. The correlation coefficient denotes a statistical relationship, and it does not imply a cause-effect relationship.

In a negative correlation, an increase in the value of one variable is accompanied by a decrease in the value of another variable, which was observed in this study. A negative correlation coefficient must be lower than −0.710 (relatively strong negative correlation) to explain more than 50% of the variance in the examined values. A relatively strong negative correlation was noted in PLT values on D1 in group ZEN5 and on D3 in groups ZEN5 and ZEN10 (see Table 6), in WBC values on D2 in groups ZEN5 and ZEN15, and in RBC values on D1 in group ZEN15, on D2 in group ZEN10, and on D3 in groups ZEN5 and ZEN15. A moderate positive correlation was observed in PLT values on D1 in group ZEN10, which could be attributed to the lowest ZEN concentration at a relatively low value of CF. These findings indicate that ZEN dose and time of exposure exert a negative effect on the haematopoietic activity of the evaluated blood morphotic components in the bone marrow microenvironment.

## 3. Discussion

In this study and in our previous research, attempts were made to fill the gap in knowledge about the in vivo effects of low and very low doses of ZEN and its metabolites on mammals [7,16,28,29]. The present study was undertaken to address the general scarcity of published data on the influence of feed contamination with ZEN and its metabolites on haematopoietic activity and the bone marrow microenvironment in pre-pubertal gilts. In this study, selective ZEN doses, in each range of their values, influenced ZEN concentrations in the bone marrow microenvironment on different dates of exposure.

### 3.1. Zearalenone and Its Metabolites in the Bone Marrow Microenvironment

The first exposure date (D1) marks the end of stimulatory processes after seven days of exposure to an undesirable substance, such as ZEN. The final effects of adaptive processes [37], accompanied by decreased haematopoiesis in the bone marrow microenvironment, are manifested on D2 [28]. These processes can be accompanied by increased Ca^2+^ levels, in particular in the mitochondria [38], and changes in the activity of selected enzymes, such as hydroxysteroid dehydrogenases [39], thus disrupting steroidogenesis [13]. Excess ZEN (which was not biotransformed or was recovered after enterohepatic recirculation) probably led to hyperestrogenism on D3. The resulting “free ZEN” can affect steroidogenesis [28].

The present findings demonstrate that the process of ZEN absorption in the bone marrow microenvironment was highly individualised, due to considerable differences in standard deviation values and specifically due to the absence of ZEN metabolites in the analysed tissue in pre-pubertal gilts. Low ZEN concentrations and the absence of ZEN metabolites in the bone marrow microenvironment on all exposure dates confirm the previous suggestion that the physiological demand for exogenous oestrogen-like compounds [28] in pre-pubertal gilts is high due to supraphysiological hormone levels [40]. However, the values noted in this study were higher than those reported in the blood serum [28] or the heart muscle [19] of the same pre-pubertal gilts. This observation could be attributed to a much higher number of oestrogen receptors in the bone marrow microenvironment than in other tissues or organs [41,42]. In menopausal females, the reproductive system, followed by the bone marrow microenvironment, are most sensitive to the presence of gonadal hormones (long-range signals) [43]. This order is reversed in maturing females (the signals are initiated by oestradiol–oestrogen receptors [44]), which could explain higher ZEN concentrations in the bone marrow environment of pre-pubertal gilts. Exposure to higher ZEN doses leads to the production of “free ZEN”, which plays different and not always positive roles. Oestradiol and “free ZEN” levels increase proportionally to the ZEN dose. These compounds decrease the concentrations of progesterone and testosterone [28], but they also play important protective roles in skeletal homeostasis [42], which is an important consideration in pre-pubertal gilts.

The above observations are confirmed by ZEN concentrations on D1 in all experimental groups. According to most researchers, metabolites are produced during the biotransformation of ZEN in pigs [45]. Therefore, the absence of metabolites in the bone marrow microenvironment could be regarded as a consequence of biotransformation processes in pre-pubertal gilts, where ZEN does not induce hyperestrogenism, but compensates for the physiological deficiency of endogenous oestrogens [40]. The above situation could result from a higher demand of the host organism for compounds with high estrogenic activity, such as ZEN and its metabolites (whose concentrations were below the sensitivity of the method) [46], or perhaps these were the physiological values that are required for the body to function [5]. This implies that it was a satisfactory effect of exposure to very low doses of ZEN during the ongoing detoxification processes. Other explanations are also possible: (i) the seventh day of exposure to an undesirable substance such as ZEN marks the end of adaptive processes [37], or (ii) ZEN is utilized as a substrate that regulates the expression of genes encoding HSDs, which act as molecular switches for the modulation of steroid hormone pre-receptors [28], or (iii) undesirable substances undergo enterohepatic circulation before they are completely eliminated from the body [47], and/or (iv) the gut microbiome exhibits specific behaviour during exposure to ZEN [29]. The above factors, alone or in combination, could affect the concentrations of ZEN in the bone marrow microenvironment, because ZEN is a promiscuous compound that can decrease the values of hematopoietic parameters [48].

On the other two exposure dates, similar correlations were noted in the average concentrations of ZEN in the bone marrow microenvironment. On these dates, ZEN concentrations were higher, but still low or very low, and ZEN metabolites were not detected. These observations could be attributed to the accumulation of ZEN (due to the saturation of active oestrogen receptors [48] and other factors that determine the concentrations of steroid hormones [45]) during prolonged exposure. However, due to the general scarcity of studies investigating the influence of low ZEN doses on the bone marrow environment and haematopoiesis, our results cannot be compared with the findings of other authors. According to the hormesis principle, exposure to very low ZEN doses affects the synthesis and secretion of sex steroid hormones. Therefore, very low ZEN doses were biotransformed in an identical manner, but the parent compound (ZEN) and, possibly, its metabolites were utilized more efficiently or completely by the macroorganism and, in particular, the bone marrow microenvironment. The interactions between endogenous and environmental (exogenous) steroids could also be influenced by other endogenous factors [49].

### 3.2. White Blood Cells

Haematological parameters somewhat exceeded the upper reference limit, in particular the percentage of lymphocytes, which increased in groups ZEN5 and ZEN10 on D1-relative lymphocytosis [50]. The increase in the percentage of lymphocytes was accompanied by a decrease in the percentage of neutrocytes. Contrary results were reported by [51], where very high ZEN concentrations exerted apoptotic effects on lymphocytes in vitro. In the present study, apoptotic effects were observed only on blood sampling date 10.

Based on the observations made by Etim et al. [52], the fact that haematological parameters were within the reference ranges indicates that feed composition had no negative effect on their values throughout the experiment. However, in view of the results of another study by Etim et al. [31], the fact that the values of WBC, NEUT, and LYMPH were within the reference ranges, but were lower than in group C, suggests that feed components did not stimulate the immune system. The neutrophil-to-lymphocyte ratio must be analysed (see Figure 3) to validate the hypothesis that the immune system was more stimulated by nutritional factors (including ZEN) and dietary stress than by functional stress [32,53]. The sizeable increase in the percentage of EOS in total WBC counts in group ZEN5, relative to group C during the entire experiment, should also be considered. In the remaining experimental groups, the percentage of EOS was lower than in group C. This observation suggests that ZEN can exert allergizing effects, which was confirmed in our previous study [54].

The observed decrease in WBC counts in all groups during the experiment points to probable immune suppression (as a consequence of gilt maturation), but this trend was least pronounced in group ZEN5. It should also be noted that during ZEN exposure, lymphocytes migrate to tissues with a physiologically higher number of oestrogen receptors, which intensifies proliferation processes and stimulates vascular filling [55]. Similar observations were made by Przybylska-Gornowicz et al. [56].

The decrease in WBC counts in the experimental groups suggests that ZEN affects adaptive processes by inducing reversible changes in the morphological and functional organization of the local immune system [50,55].

### 3.3. Red Blood Cells

Red blood cell counts remained stable throughout the experiment. Considerable differences were noted on blood sampling dates 4 and 10, with a rising trend in group C. In the experimental groups, a decrease in Fe levels [7] had no effect on RBC counts. Haemoglobin distribution width values were stable, but very low, and they were accompanied by a drop in glucose levels [7]. The above values are characteristic of pre-pubertal animals [57]. 

Erythrocytes, the main morphotic components of the blood, are highly sensitive to endogenous factors such as oxidation. Red blood cells have effective antioxidant mechanisms that remove reactive oxygen species that are produced outside or inside erythrocytes [58]. According to Tatay et al. [59], ZEN is a powerful antioxidant, and one of the side effects of ZEN exposure is a decrease in RBC counts, which could be attributed to “antioxidant competition”. Similar suggestions were made by other authors who reported that ZEN induced eryptosis in mature human erythrocytes [60]. According to the cited authors, eryptosis can take place even at low concentrations of ZEN.

Eryptosis can also be induced by a mechanistic increase in intracellular Ca^2+^ levels [61,62], oxidative stress, energy depletion [63], and exposure to undesirable substances such as ZEN [56]. However, the loss of extracellular Ca^2+^ can also inhibit the eryptotic effect [64] due to anaemia and microcirculatory disorders. Microcirculatory disorders are observed in various tissues during exposure to ZEN, and they lead to numerous extravasations and vasodilation [65,66] in pre-pubertal female mammals, due to the relaxation of smooth muscle cells in vessel walls, in particular in large veins, large arteries, and smaller arterioles. An analysis of RBC values clearly indicates that all of the tested ZEN doses decreased erythrocyte counts and induced eryptosis [41].

### 3.4. Platelets

Platelet counts were above the norm in all groups, which is indicative of thrombocytosis [67]. The percentage of PLT Clumps was also elevated (pseudothrombocytopenia). These results could be explained by inadequate analytical standards or blood artefacts [68]. It should also be noted that platelets participate in the early stages of liver regeneration, but the underlying mechanisms of action have not yet been fully elucidated [69]. The liver is the main organ responsible for the biotransformation (detoxification) of undesirable substances such as ZEN [13]. The observed differences in values can be attributed to: (i) intensified proliferation in group ZEN5, and (ii) the influence of substances that stimulate proliferation in maturing animals [7,67,70].

In an analysis of PLT values, a positive correlation coefficient was noted only once. On the remaining exposure dates, the correlation coefficient assumed negative values, and it revealed the presence of a relatively strong negative correlation on three occasions. The strongest negative correlations were noted on D1 (during the highest exposure to ZEN) and D3 (during prolonged hyperestrogenism resulting from the accumulation of ZEN).

### 3.5. Summary and Conclusions

The exposure to very low doses of ZEN, administered *per os* to pre-pubertal gilts over a period of 42 days, induced a specific response in the bone marrow microenvironment. Zearalenone metabolites were not detected, and the concentration of the parent compound was higher in the bone marrow microenvironment than in other tissues. Low doses of ZEN affected haematopoiesis and induced completely different responses in selected haematological parameters than high doses (LOAEL). Zearalenone’s multidirectional effects on blood morphotic components at the beginning and end of the experiment were probably modified by compensatory and adaptive mechanisms. Zearalenone also induced reversible changes in the morphological and functional organization of the local immune system. The values of the analysed haematological parameters pointed to eryptosis and the stimulation of detoxifying organs such as the liver, where the intensity of biotransformation processes decreases under exposure to low doses of the studied mycotoxin. Further research is needed to elucidate the mechanisms responsible for changes in the haematological parameters of pre-pubertal gilts exposed to low doses of ZEN.

The results of these observations should play an important role in clinical veterinary practice, where ZEN could become a biomarker of the subclinical state of pre-pubertal gilts.

## 4. Materials and Methods

### 4.1. General Information

All experimental procedures involving animals were carried out in compliance with Polish regulations setting forth the terms and conditions of animal experimentation (Opinions No. 12/2016 and 45/2016/DLZ of the Local Ethics Committee for Animal Experimentation of 27 April 2016 and 30 November 2016).

### 4.2. Experimental Feed

Analytical samples of ZEN were dissolved in 96 µL of 96% ethanol (SWW 2442-90, Polskie Odczynniki SA, Katowice, Poland) in doses appropriate for different BW. Feed was placed in gel capsules saturated with the solution and kept at room temperature to evaporate the alcohol. All groups of gilts received the same feed throughout the experiment. The animals were weighed at weekly intervals, and the results were used to adjust individual mycotoxin doses [12,16,29].

The feed administered to all experimental animals was supplied by the same producer. Friable feed was provided *ad libitum* twice daily, at 8:00 a.m. and 5:00 p.m., throughout the experiment. The composition of the complete diet, as declared by the manufacturer, is presented in Table 7 [12,16,29].

The proximate chemical composition of the diets fed to pigs in groups C, ZEN5, ZEN10, and ZEN15 was determined using the NIRS™ DS2500 F feed analyser (FOSS, Hillerød, Denmark), a monochromator-based NIR reflectance and transflectance analyser with a scanning range of 850–2500 nm [12,16,29].

#### Toxicological Analysis in Feed

Feed was analysed for the presence of mycotoxins and their metabolites: ZEN, α-ZEL, and deoxynivalenol (DON). Mycotoxin concentrations in feed were determined by separation in immunoaffinity columns (Zearala-Test^TM^ Zearalenone Testing System, G1012, VICAM, Watertown, MA, USA; DON-Test^TM^ DON Testing System, VICAM, Watertown, MA, USA). The feed samples were ground in a laboratory mill. 25 g of the ground sample was taken up in 150 mL of acetonitrile (90%) to extract the mycotoxins. 10 mL of the resulting solution was withdrawn and diluted with 40 mL of water. Again, 10 mL of the next solution was taken and passed through the immunoaffinity column (VICAM). The immunoaffinity bed in the column was subsequently washed with demineralized water (Millipore Water Purification System, Millipore S.A., Molsheim, France). The column was eluted with 99.8% methanol (LIChrosolvTM, No. 1.06 007, Merck-Hitachi, München, Germany) to wash away the bound mycotoxin. The obtained solutions were analyzed by high-performance liquid chromatography (HPLC system, Hewlett Packard type 1050 and 1100, Beeston Court, United Kingdom) coupled with a diode array detector (DAD) and a fluorescence detector (FLD) and chromatography columns (Atlantis T3, 3 µm 3.0 150 mm Column No. 186003723, Waters, AN Etten-Leur, Ireland). Mycotoxins were separated using a mobile phase of acetonitrile:water:methanol (46:46:8, *v*/*v*/*v*). The flow rate was 0.4 mL/min. The limit of detection was set at 5 μg/kg of feed for DON and 2 μg/kg of feed for ZEN, based on validation of chromatographic methods for the determination of ZEN and DON levels in feed materials and feeds. Chromatographic methods were validated at the Department of Veterinary Prevention and Feed Hygiene, Faculty of Veterinary Medicine, University of Warmia and Mazury in Olsztyn, Poland [11].

### 4.3. Experimental Animals

The experiment was performed at the Department of Veterinary Prevention and Feed Hygiene of the Faculty of Veterinary Medicine at the University of Warmia and Mazury in Olsztyn Poland on 60 clinically healthy pre-pubertal gilts with initial BW of 14.5 ± 2 kg. The animals were housed in the same building, in adjoining pens with latticework walls, and had free access to water. They were randomly assigned to three experimental groups (group ZEN5, group ZEN10, and group ZEN15; *n* = 15) and a control group (group C; *n* = 15) [71,72]. Group ZEN5 gilts were orally administered ZEN at 5 μg/kg BW, Group ZEN10, 10 μg/kg BW, and Group ZEN15, 15 μg/kg BW. Group C animals were orally administered a placebo. Zearalenone was administered daily in gel capsules before morning feeding. Feed was the carrier, and group C pigs were administered the same gel capsules, but without mycotoxins [12,16,29].

#### 4.3.1. Toxicological Analysis of Bone Marrow Microenvironment

##### Tissues Samples

Five prepubertal gilts from every group were euthanized on analytical date 1 (D1, exposure day 7), date 2 (D2, exposure day 21), and date 3 (D3, exposure day 42) by intravenous administration of pentobarbital sodium (Fatro, Ozzano Emilia, Bologna, Italy) and bleeding. Samples were taken from the iliac wing (posterior superior iliac spine) immediately after cardiac arrest and were rinsed with phosphate buffer. The collected samples were stored at a temperature of −20 °C.

##### Extraction Procedure

The presence of ZEN, α-ZEL, and β-ZEL in the bone marrow microenvironment was determined with the use of immunoaffinity columns. The samples of bone marrow environment (3 mL) were transferred to centrifuge tubes homogenized with 7 mL of methanol (99.8%) for 4 min. The tubes were vortexed 4 times at 5 min intervals, after which they were centrifugated at 5000 rpm for 15 min. 5 mL was taken from the suspension obtained and taken up in 20 mL of deionized water, and only from this solution, 12.5 mL was taken for zearalenone extraction. The supernatant was carefully collected and passed through immunoaffinity columns (Zearala-TestTM Zearalenone Testing System, G1012, VICAM, Watertown, MA, USA) at the rate of 1–2 drops per second. The immunoaffinity bed in the column was subsequently washed with demineralized water (Millipore Water Purification System, Millipore S.A., Molsheim, France). The column was eluted isocratic with 99.8% methanol (LIChrosolvTM, No. 1.06 007, Merck-Hitachi, Germany) to wash away the bound mycotoxin. After extraction, the eluents were placed in a water bath at a temperature of 50 °C and were evaporated in a stream of nitrogen. Dry residues were stored at −20 °C until chromatographic analysis. Next, 0.5 mL of 99.8% acetonitrile (ACN) was added to dry residues to dissolve the mycotoxin. The process was monitored with the use of internal standards (Cayman Chemical 1180 East Ellsworth Road Ann Arbor, MI 48108, USA, ZEN-catalog number 11353; Batch 0593470-1; *a*-ZEN-catalog number 16549; Batch 0585633-2; *β*-ZEN-catalog number 19460; Batch 0604066-7), and the results were validated by mass spectrometry.

##### Chromatographic Quantification of ZEN and Its Metabolites

Zearalenone and its metabolites were quantified at the Institute of Dairy Industry Innovation in Mrągowo, Poland. The biological activity of ZEN, α-ZEL, and β-ZEL in the bone marrow microenvironment was determined by combined separation methods, involving immunoaffinity columns (Zearala-TestTM Zearalenone Testing System, G1012, VICAM, Watertown, MA, USA), Agilent 1260 liquid chromatography (LC) system, and a mass spectrometry (MS, Agilent 6470, Santa Clara, United States) system. Samples were analyzed on a chromatographic column (Atlantis T3, 3 μm 3.0 × 150 mm, column No. 186003723, Waters, AN Etten-Leur, Ireland). The mobile phase was composed of 70% acetonitrile (LiChrosolvTM, No. 984 730 109, Merck-Hitachi, Mannheim, Germany), 20% methanol (LiChrosolvTM, No. 1.06 007, Merck-Hitachi, Mannheim, Germany), and 10% deionized water (MiliporeWater Purification System, Millipore S.A. Molsheim, France) with the addition of 2 mL of acetic acid per 1 L of the mixture. The flow rate was 0.4 mL/min, and the temperature of the oven column was 40 °C. The chromatographic analysis was completed in 4 min. The column was flushed with 99.8% methanol (LIChrosolvTM, No. 1.06 007, Merck-Hitachi, Mannheim, Germany) to remove the bound mycotoxin. The flow rate was 0.4 mL/min, and the temperature of the oven column was 40 °C. The chromatographic analysis was completed in 4 min.

Mycotoxin concentrations were determined with an external standard and were expressed in ppb (ng/mL). Matrix-matched calibration standards were applied in the quantification process to eliminate matrix effects that can decrease sensitivity. Calibration standards were dissolved in matrix samples based on the procedure that was used to prepare the remaining samples. The material for calibration standards was free of mycotoxins. The limits of detection (LOD) for ZEN, α-ZEL, and β-ZEL were determined as the concentration at which the signal-to-noise ratio decreased to 3. The concentrations of ZEN, α-ZEL, and β-ZEL were determined in each group and on three analytical dates (see Table 1).

##### Carryover Factor

Carryover toxicity takes place when an organism is able to survive under exposure to low doses of mycotoxins. Mycotoxins can compromise tissue or organ functions [73] and modify their biological activity [7,28]. The CF was determined in the bone marrow microenvironment when the daily dose of ZEN (5 µg ZEN/kg BW, 10 µg ZEN/kg BW, or 15 µg ZEN/kg BW) administered to each animal was equivalent to 560–32251.5 µg ZEN/kg of the complete diet, depending on daily feed intake. Mycotoxin concentrations in tissues were expressed in terms of the dry matter content of the samples.

The CF was calculated as follows:CF = toxin concentration in tissue [ng/g]/toxin concentration in diet [ng/g].

##### Statistical Analysis

Data were processed statistically at the Department of Discrete Mathematics and Theoretical Computer Science, Faculty of Mathematics and Computer Science of the University of Warmia and Mazury in Olsztyn, Poland. The bioavailability of ZEN and its metabolites in the bone marrow microenvironment was analysed in group C and three experimental groups on three analytical dates. The results were expressed as means (±) with standard deviation (SD). The following parameters were analysed: (i) differences in the mean values for three ZEN doses (experimental groups) and the control group on both analytical dates, and (ii) differences in the mean values for specific ZEN doses (groups) on both analytical dates. In both cases, the differences between mean values were determined by one-way ANOVA. If significant differences were noted between groups, the differences between paired means were determined by Tukey’s multiple comparison test. If all values were below LOD (mean and variance equal zero) in any group, the values in the remaining groups were analysed by one-way ANOVA (if the number of the remaining groups was higher than two), and the means in these groups were compared against zero by Student’s t-test. Differences between groups were determined by Student’s t-test. The results were regarded as highly significant at *p* < 0.01 (**) and as significant at 0.01 < *p*< 0.05 (*). Data were processed statistically using Statistica v.13 software (TIBCO Software Inc., Silicon Valley, CA, USA, 2017). Dose–response relationships were determined by Pearson’s correlation analysis. Differences were regarded as significant at *p* ≤ 0.05. The results were presented as means ± standard error of the mean (S.E.M.).

### 4.4. Blood Sampling for Metabolic Profile Analysis

Blood for haematology tests was sampled from the *vena cava cranialis* ten times: on the first day (first date) of the experiment and on nine successive dates. Blood was sampled within 20 s after the immobilization of pre-pubertal gilts [74]. Blood was sampled from 5 gilts from every group on each sampling date.

### 4.5. Haematology Tests

Blood samples of 2 mL were collected from pre-pubertal gilts into test tubes containing EDTAK_2_ (Ethylenediaminetetraacetic acid dipotassium salt dihydrate) (Sigma Aldrich, Darmstadt, Germany) as anticoagulant. The samples were thoroughly mixed and analysed to determine: Red Blood Cell (RBC) counts, Mean Corpuscular Volume (MCV), Mean Corpuscular Haemoglobin Concentrations (MCHC), Mean Corpuscular Haemoglobin (MCH), Haematocrit (HCT), and White Blood Cell (WBC) counts in a Medonic (Siemens, Plantation, FL, USA) haematology analyser according to the procedure recommended by the manufacturer. Measurements were performed in K2-EDTA whole blood by laser flow cytometry in the Siemens Advia 2120I (Erlangen, Germany) haematology analyser equipped with: (i) optical peroxide biosensor which measures dispersed light and light absorbed by individual cells by hydrodynamic focusing on a cell stream in a flow-through cuvette, (ii) laser optics for measuring high-angular and low-angular light dispersion and absorption by individual cells, where the laser diode was the source of light; the measurement was performed to evaluate red blood cells, platelets, and lobulation of nuclei in white blood cells, (iii) HGB (Haemoglobin) colorimeter, for measuring lamp voltage corresponding to the amount of transmitted light, and (iv) PEROX and BASO leukocytes cytograms (ADVIA 2120i -Siemens Healthcare®, Bayer, Germany) reagents, for generating differential cytograms.

Randomly selected samples were analysed in two replications. Repeatable results were obtained.

### 4.6. Statistical Analysis

The results were grouped based on: (i) duration of the experiment in group C and the experimental groups on a given sampling date, and (ii) sampling dates for a given parameter. The results were processed in the Statistica application (Statistica 9.0, StatSoft Kraków, Poland). Differences between groups (parameter or sampling date) were determined by ANOVA. The use of ANOVA was justified by the Brown-Forsythe test for the equality of group variances. When differences between groups were statistically significant (*p* < 0.01, highly significant differences; 0.01 < *p* < 0.05, significant differences; *p* > 0.05, no differences), Tukey’s HSD test was used to identify the groups that were significantly different. Linear correlations between the concentrations of ZEN in bone marrow environment in fixed groups were determined based on the values of the Pearson’s correlation coefficient [75]. Data were processed in Statistica v. 13 (TIBCO Software Inc., Silicon Valley, CA, USA, 2017).

## Figures and Tables

**Figure 1 toxins-14-00105-f001:**
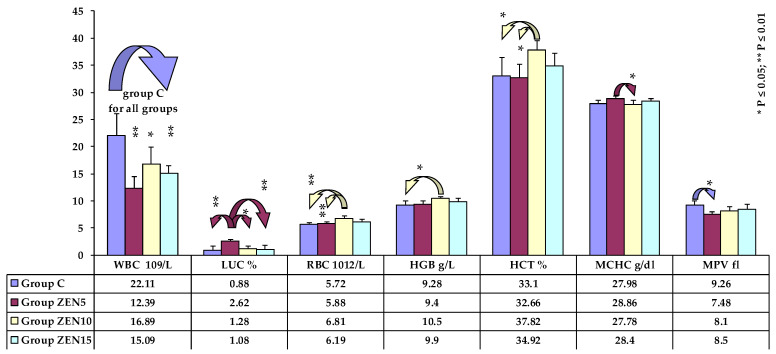
Selected haematological parameters on the fourth analytical date ($\bar{x}$, SD) Key: C, control group; Group ZEN5, 5 μg ZEN/kg BW; Group ZEN10, 10 μg ZEN/kg BW; Group ZEN15, 15 μg ZEN/kg BW. Statistically significant differences: * at *p* ≤ 0.05; ** at *p* ≤ 0.01.

**Figure 2 toxins-14-00105-f002:**
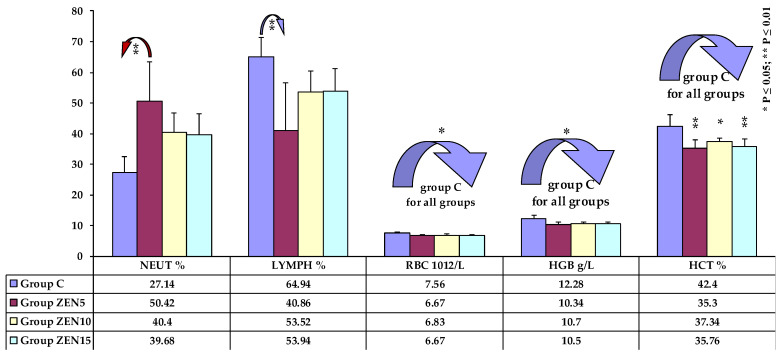
Selected haematological parameters on the tenth analytical date ($\bar{x}$, SD). Key: C, control group; Group ZEN5, 5 μg ZEN/kg BW; Group ZEN10, 10 μg ZEN/kg BW; Group ZEN15, 15 μg ZEN/kg BW. Statistically significant differences: * at *p* ≤ 0.05; ** at *p* ≤ 0.01.

**Figure 3 toxins-14-00105-f003:**
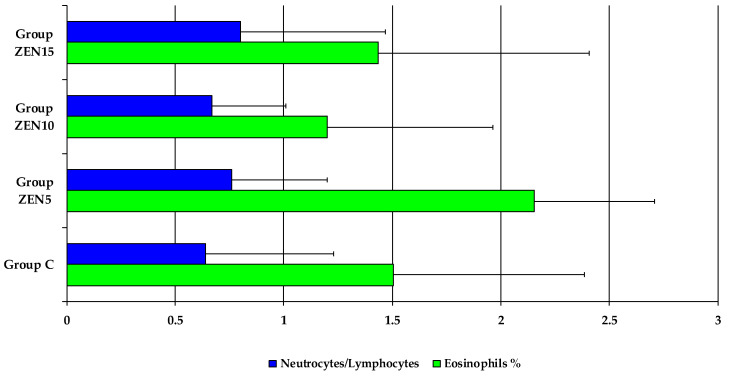
Selected haematological parameters in groups ($\bar{x}$). Key: C, control group; Group ZEN5, 5 μg ZEN/kg BW; Group ZEN10, 10 μg ZEN/kg BW; Group ZEN15, 15 μg ZEN/kg BW.

**Table 1 toxins-14-00105-t001:** The carry-over factor (CF) and mean ($\bar{x}$) concentrations of ZEN and ZELs (ng/g) in the bone marrow microenvironment of pre-pubertal gilts.

Exposure Date	Feed Intake [kg/Day]	Total ZEN Doses in Groups Respectively [µg/kg BW]	Group ZEN5 [ng/g]	CF Group ZEN5	Group ZEN10 [ng/g]	CF Group ZEN10	Group ZEN15 [ng/g]	CF Group ZEN15
**Zearalenone**								
D1	0.8	80.5/161.9/242.7	5.64 ± 3.24	7 × 10^−5^	6.35 ± 3.56	39 × 10^−6^	8.17 ± 0.45	33 × 10^−6^
D2	1.1	101.01/196.9/298.2	4.69 ± 4.28	46 × 10^−6^	7.08 ± 0.10	35 × 10^−6^	6.84 ± 0.18 ^●●^	22 × 10^−6^
D3	1.6	128.3/481.4/716.7	7.74 ± 0.26	6 × 10^−5^	7.35 ± 0.25 *	15 × 10^−6^	7.03 ± 0.14 **^,^^●●^	98 × 10^−7^
**α-ZEL and β-ZEL**								
D1–D3	not applicable		0					

**Abbreviation**: D1, exposure day 7; D2, exposure day 21; D3, exposure day 42. Experimental groups: Group ZEN5, 5 µg ZEN/kg BW; Group ZEN10, 10 µg ZEN/kg BW; Group ZEN15, 15 µg ZEN/kg BW. LOD > values below the limit of detection were expressed as 0. Statistically significant differences were determined at * *p* ≤ 0.05 and **^,●●^
*p* ≤ 0.01; *, ** statistical difference between group ZEN5 vs. group ZEN10 and group ZEN15 on exposure date D3; ^●●^ statistical difference in group 3 between exposure date D1 and exposure dates D2 and D3.

**Table 2 toxins-14-00105-t002:** Selected haematological parameters in group ZEN5 on different analytical dates ($\bar{x}$, SD).

Blood Collection Dates	WBC 10^9^/L	MONO %	LUC %	MPV fl
1	15.52 ± 7.58	5.32 ± 1.06	0.88 ± 0.63 ^bb^	7.36 ± 1.61 ^cc,ff^
2	18.8 ± 4.84	4.94 ± 1.07	1.72 ± 0.99	8.94 ± 0.99 ^a^
3	18.8 ± 2.71	6.04 ± 3.69	1.42 ± 0.94	7.68 ± 0.55 ^c,^^f^
4	12.39 ± 2.1 ^ee^	4.3 ± 1.02	2.62 ± 0.31	7.48 ± 0.54 ^c,^^ff^
5	17.23 ± 0.57	3.56 ± 0.55	1.26 ± 0.38	9.44 ± 1.44
6	15.45 ± 1.41	6.78 ± 0.91	0.84 ± 0.76 ^bb^	8.14 ± 0.59
7	12.46 ± 1.37 ^ee^	3.84 ± 0.89	1.44 ± 0.73	8.44 ± 0.87
8	15.04 ± 10.37	6.4 ± 2.59	0.8 ± 0.53 ^b^	8.82 ± 1.42
9	24.55 ± 5.39	5.78 ± 1.18	0.6 ± 0.35 ^bb^	8.82 ± 1.35
10	10.44 ± 3 ^ee^	2.82 ± 0.84 ^d^	0.72 ± 0.37 ^bb^	9.66 ± 1.88

Key: Group ZEN5, 5 μg ZEN/kg BW; WBC, White Blood Cells; MONO, Monocytes; LUC, Large Unstained Cells; MPV, Mean Platelet Volume. Statistical symbols: ^a^, relative to date 1; ^b^, relative to date 4; ^c^, relative to date 5; ^d^, relative to date 6; ^e^, relative to date 9; ^f^, relative to date 10. Statistically significant differences: ^a,b,c,d^ and ^f^ at *p* ≤ 0.05; ^bb,cc,^^ee^ and ^ff^ at *p* ≤ 0.01.

**Table 3 toxins-14-00105-t003:** Selected white blood cell markers in group ZEN10 on different analytical dates ($\bar{x}$, SD).

Blood Collection Dates	WBC 10^9^/L	NEUT %	LYMPH %	LUC %	RBC 10^12^/L	HGB g/L	HCT %	MCV fl	MCHC g/dL	HDW g/dL
1	21.86 ± 2.67 ^d^	37.68 ± 3.89	55.7 ± 3.89	0.8 ± 0.64	6.68 ± 0.54	10.79 ± 0.65	37.08 ± 2.19	55.6 ± 1.81	29.1 ± 1.23	1.58 ± 0.13 ^aa,bb^
2	22.74 ± 3.6 ^a,^^d^	43.7 ± 12.26 ^a^	50.02 ± 12.12 ^a^	0.86 ± 0.69	6.57 ± 0.38 ^c^	10.22 ± 0.36 ^b,cc^	34.98 ± 1.27 ^bb,cc^	53.3 ± 2.52	29.12 ± 0.57 ^a^	1.57 ± 0.04 ^a,b^
3	19.55 ± 3.03	45.14 ± 5.14 ^a^	46.94 ± 3.13 ^aa^	0.98 ± 0.62	6.56 ± 0.17 ^c^	10.18 ± 0.34 ^b,^^cc^	35.48 ± 0.74 ^bb,cc^	54.14 ± 2.49	28.7 ± 0.51	1.6 ± 0.08 ^a,bb^
4	16.89 ± 2.94	29.58 ± 7.88	63.9 ± 7.68	1.28 ± 0.37	6.81 ± 0.39	10.5 ± 0.36	37.82 ± 1.63	55.6 ± 1.75	27.78 ± 0.8	1.54 ± 0.09 ^a^
5	22.6 ± 6.49 ^a,d^	43.34 ± 11.1	48.62 ± 10.43 ^a^	1.86 ± 0.34 ^c^	6.83 ± 0.46	10.28 ± 0.76 ^b,^^c^	35.7 ± 2.87 ^bb,cc^	52.22 ± 1.6 ^a^	28.72 ± 0.38	1.54 ± 0.05 ^a^
6	15.36 ± 3.68	33.7 ± 6.15	57.92 ± 6.89	1.22 ± 0.61	6.71 ± 0.33	10.62 ± 0.97	38.06 ± 2.54	56.68 ± 1.98	27.86 ± 0.86	1.55 ± 0.07 ^a,b^
7	13.59 ± 2.68	27.28 ± 8.23	66.96 ± 8.07	0.66 ± 0.45	6.51 ± 0.42 ^b,^^c^	10.34 ± 0.28 ^c^	37.56 ± 1.11	57.86 ± 2.94	27.48 ± 0.54	1.43 ± 0.06
8	15.43 ± 5.68	34.08 ± 5.43	58.78 ± 5.47	1.64 ± 0.01	7.35 ± 0.33	11.54 ± 0.6	40.54 ± 2.56	55.18 ± 3.32	28.44 ± 0.6	1.44 ± 0.05
9	19.81 ± 3.64	39.6 ± 5.69	53.58 ± 6.26	0.52 ± 0.15	7.47 ± 0.41	11.7 ± 0.43	40.92 ± 0.86	54.84 ± 2.3	28.54 ± 0.94	1.53 ± 0.06 ^a^
10	12.85 ± 3.75	40.4 ± 6.31	53.52 ± 6.83	0.88 ± 0.21	6.83 ± 0.4	10.7 ± 0.55	37.34 ± 1.23	54.74 ± 2.65	28.62 ± 0.79	1.54 ± 0.08 ^a^

Key: Group ZEN10, 10 μg ZEN/kg BW; WBC, White Blood Cells; NEUT, Neutrophils; LYMPH, Lymphocytes; LUC, Large Unstained Cells; RBC, Red Blood Cells; HGB, Haemoglobin; HCT, Haematocrit; MCV, Mean Corpuscular Volume; MCHC, Mean Corpuscular Haemoglobin Concentrations; HDW, Haemoglobin Distribution Width. Statistical symbols: ^a^, relative to date 7; ^b^, relative to date 8; ^c^, relative to date 9; ^d^, relative to date 10. Statistically significant differences: ^a^^,^^b^^,^^c^ and ^d^ at *p* ≤ 0.05; ^aa^^,^^bb^ and ^cc^ at *p* ≤ 0.01.

**Table 4 toxins-14-00105-t004:** Selected haematological parameters in group ZEN15 on different analytical dates ($\bar{x}$, SD).

Blood Collection Dates	WBC 10^9^/L	NEUT %	LYMPH %	EOS %	MPV fl
1	22.01 ± 3.85 ^ee,ff^	52.92 ± 7.91	39.96 ± 8.22 ^e^	2.4 ± 1.47	8.22 ± 1.65
2	19.83 ± 1.35 ^e,f^	43.7 ± 6.64	48.62 ± 5.62	1.32 ± 0.52	7.68 ± 1.55 ^a^
3	20.73 ± 2.21 ^e,f^	44.92 ± 7.66	46.42 ± 6.85	1.96 ± 0.65	7.50 ± 0.45
4	15.09 ± 14.43	38.54 ± 7.92	55.38 ± 7.23	0.64 ± 0.27 ^a^	8.51 ± 1.12
5	22.33 ± 5.96 ^ee,ff^	48.16 ± 12.12	45.62 ± 10.12	1.22 ± 0.43	7.58 ± 1.05 ^aa,^^b,^^c^
6	16.66 ± 3.96	37 ± 5.58	54.14 ± 5.95	1.6 ± 0.41	7.84 ± 1.09
7	10.79 ± 2.53	26.44 ± 6.28 ^a,d^	66.12 ± 4.45	0.74 ± 0.23 ^a^	7.44 ± 1.32
8	19.04 ± 2.47	41.32 ± 11.48	49.64 ± 11.23	1.9 ± 0.76	8.38 ± 2.72
9	14.8 ± 10.25	29.02 ± 20.07 ^a^	40.37 ± 27.41 ^e^	1.15 ± 0.82	8.85 ± 2.46
10	10.73 ± 1.78	39.68 ± 6.79	53.94 ± 7.3	1.44 ± 1.05	8.28 ± 1.05 ^aa,^^b,^^cc^

Key: Group ZEN15, 15 μg ZEN/kg BW; WBC, White Blood Cells; NEUT, Neutrophils; LYMPH, Lymphocytes; EOS, Eosinophils; MPV, Mean Platelet Volume. Statistical symbols: ^a^, relative to date 1; ^b^, relative to date 3; ^c^, relative to date 4; ^d^, relative to date 5; ^e^, relative to date 7; ^f^, relative to date 10. Statistically significant differences: ^a,^^b^^,c,d,e^ and ^f^ at *p* ≤ 0.05; ^aa,cc,ee^ and ^ff^ at *p* ≤ 0.01.

**Table 5 toxins-14-00105-t005:** Platelet count in the experimental groups on different analytical dates ($\bar{x}$, SD).

Blood Collection Dates	PLT 10^9^/L Group ZEN5	PLT 10^9^/L Group ZEN10	PLT Clumps % Group ZEN10	PLT 10^9^/L Group ZEN15	PLT Clumps % Group ZEN15
1	405.4 ± 295.39	690.2 ± 121.87 ^e^	26.4 ± 14.75	479.8 ± 214.68	19.8 ± 18.07
2	522.6 ± 112.81	763 ± 107.01 ^d,ee^	19.8 ± 18.07	668 ± 144.82 ^d,e^	19.8 ± 18.07
3	745 ± 54.41 ^e^	813.2 ± 149.05 ^c,dd,ee^	0 ± 0	753.8 ± 167.69 ^a,dd,^^ee^	0 ± 0
4	741.4 ± 89.6 ^e^	734.6 ± 131.75 ^d,e^	26.4 ± 14.75	501.6 ± 62.13 ^b^	33 ± 0 ^b^
5	517.2 ± 162.41	800.2 ± 212.97 ^c,dd,ee^	33 ± 0 ^bb^	718.2 ± 129.56 ^a,d,ee^	26.4 ± 14.75
6	645.2 ± 140.47	650.6 ± 238.35	19.8 ± 18.07	635.4 ± 38.01 ^e^	19.8 ± 18.07
7	463.6 ± 83.09	569.4 ± 132.14	26.4 ± 14.75	607.8 ± 203.15	13.2 ± 18.07
8	454 ± 317.27	617.2 ± 180.88	13.2 ± 18.07	626.6 ± 263.92 ^e^	19.8 ± 18.07
9	595.6 ± 172.33	497.2 ± 101.9	33 ± 0 ^bb^	394 ± 294.02	24.75 ± 16.5
10	369.4 ± 118.85	478.2 ± 151.9	33 ± 0 ^bb^	386.6 ± 158.7	33 ± 0 ^b^

Key: PLT count in groups 1, 2, and 3; PLT clumps in groups 2 and 3. Statistical symbols: ^a^, relative to date 1; ^b^, relative to date 3; ^c^, relative to date 7; ^d^, relative to date 9; ^e^, relative to date 10. Statistically significant differences: ^a,^^b^^,c,^^d^ and ^e^ at *p* ≤ 0.05; ^bb^^,^^dd^ and ^ee^ at *p* ≤ 0.01.

**Table 6 toxins-14-00105-t006:** Correlation coefficients (Pearson’s *r*).

Exposure Date	Experimental Groups	PLT	WBC	RBC
D1	Group ZEN5	−0.739	−0.362	−0.456
	Group ZEN10	0.685	−0.502	−0.189
	Group ZEN15	−0.590	−0.336	−0.869
D2	Group ZEN5	−0.251	−0.851	−0.424
	Group ZEN10	−0.559	−0.325	−0.832
	Group ZEN15	−0.682	−0.797	−0.565
D3	Group ZEN5	−0.772	−0.230	−0.752
	Group ZEN10	−0.731	−0.451	−0.300
	Group ZEN15	−0.436	−0.128	−0.738

Key: Ratio of ZEN concentrations in the bone marrow microenvironment and blood morphotic components (RBC, WBC, and PLT) on different exposure dates (D1, D2, and D3: exposure days 7, 21, and 42, respectively). The mycotoxin was administered once daily before morning feeding (group ZEN5, 5 μg ZEN/kg BW; group ZEN10, 10 μg ZEN/kg BW; group ZEN15, 15 μg ZEN/kg BW). Samples were collected from pre-pubertal gilts immediately before slaughter.

**Table 7 toxins-14-00105-t007:** Declared composition of the complete diet.

Parameters	Composition Declared by the Manufacturer (%)
Soybean meal	16
Wheat	55
Barley	22
Wheat bran	4.0
Chalk	0.3
Zitrosan	0.2
Vitamin-mineral premix ^1^	2.5

^1^ Composition of the vitamin-mineral premix per kg: vitamin A, 500.000 IU; iron, 5000 mg; vitamin D3, 100.000 IU; zinc, 5000 mg; vitamin E (alpha-tocopherol), 2000 mg; manganese, 3000 mg; vitamin K, 150 mg; copper (CuSO_4_·5H_2_O), 500 mg; vitamin B_1_, 100 mg; cobalt, 20 mg; vitamin B_2_, 300 mg; iodine, 40 mg; vitamin B_6_, 150 mg; selenium, 15 mg; vitamin B_12_, 1500 μg; L-lysine, 9.4 g; niacin, 1200 mg; DL-methionine+cystine, 3.7 g; pantothenic acid, 600 mg; L-threonine, 2.3 g; folic acid, 50 mg; tryptophan, 1.1 g; biotin, 7500 μg; phytase+choline, 10 g; ToyoCerin probiotic+calcium, 250 g; antioxidant+mineral phosphorus and released phosphorus, 60 g; magnesium, 5 g; sodium and calcium, 51 g.

## Data Availability

Not applicable.

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
