# Peer review of "The Effect of Low Doses of Zearalenone (ZEN) on the Bone Marrow Microenvironment and Haematological Parameters of Blood Plasma in Pre-Pubertal Gilts"

_toxins, 2022, doi:10.3390/toxins14020105_

Round 1
Reviewer 1 Report
In this study the effect of low dose ZEN was studied in pre-pubertal gilts. The authors conclude that the administration of ZEN resulted in the change of haematological parameters. The topic has great relevance because the low dose effect of mycotoxins is poorly studied. The manuscript contains valuable results. However the manuscript is hard to follow and needs thorough revision. My comments:
- It would be better to start with the information about ZEN instead of bone marrow in the introduction section since understanding ZEN function is the aim of the study.
- It is hard to understand the experimental progress and interpret the data in the results section thus it needs thorough revision as follows.
- It is necessary to summarize the experimental design in the beginning of this section.
- In the section 2.1 it is written that “The analysed feed did not contain mycotoxins, or its mycotoxin content was below the sensitivity of the method”. Where are the data that prove this?
- The same concern is for the 2.2 section about “clinical observations”. If you refer to your previous studies only do not separate this observation in a separated section.
- What is presented in Table 2? The results of the control group? Why is it necessary to present the control results in a separate table? It would be more understandable to present the effect of ZEN administration only relative to the control samples. On the other hand the symbols used to present significance are not conventional. It would be better to use e.g. a,b,c letters instead.
- Regarding Figure 1 and 2 it would be enough to use the data of the tables only, the figure is not necessary. Regarding the statistics comparing the results of ZEN groups with the control group is the most relevant in order to see the effect of ZEN administration. In the present form of the table this relevance is hard to interpret. Please repeat the statistical analysis in the case of MCHC and HGB (in Figure 1) and in the case of RBC (in Figure 2).
- Regarding Table 3, 4 and 5 it would be more understandable to use figures instead of tables. Why not the same parameters are presented in these tables? If you want to compare the effect of ZEN administration in different doses the same parameters need to be presented.
- It is not necessary to present what correlation means in the section 2.6. However, it is required to present the experimental design and to explain which factors were chosen to see correlations. Only those that are presented in Table 7? Why these?
- The discussion section should provide an explanation for the observed results. In the present form it repeats the results and contain a lot of information that should be relocate to the results section including Figure 3 that would make the results more understandable. Only the explanation of the data should remain in this section.
- What is the practical importance of your data? E.g. in breeding animals? It should be discussed in the conclusion section.
Author Response
Open Review
(x) I would not like to sign my review report
( ) I would like to sign my review report
English language and style
( ) Extensive editing of English language and style required
( ) Moderate English changes required
( ) English language and style are fine/minor spell check required
(x) I don't feel qualified to judge about the English language and style
|
Yes |
Can be improved |
Must be improved |
Not applicable |
|
|
Does the introduction provide sufficient background and include all relevant references? |
( ) |
(x) |
( ) |
( ) |
|
Is the research design appropriate? |
( ) |
(x) |
( ) |
( ) |
|
Are the methods adequately described? |
( ) |
( ) |
(x) |
( ) |
|
Are the results clearly presented? |
( ) |
( ) |
(x) |
( ) |
|
Are the conclusions supported by the results? |
( ) |
(x) |
( ) |
( ) |
Comments and Suggestions for Authors
In this study the effect of low dose ZEN was studied in pre-pubertal gilts. The authors conclude that the administration of ZEN resulted in the change of haematological parameters. The topic has great relevance because the low dose effect of mycotoxins is poorly studied. The manuscript contains valuable results. However the manuscript is hard to follow and needs thorough revision. My comments:
Dear Reviewer - all comments and changes in the text, are presented in red.
It would be better to start with the information about ZEN instead of bone marrow in the introduction section since understanding ZEN function is the aim of the study.
Dear Reviewer - the introduction has been redrafted in line with the suggestions presented.
It is hard to understand the experimental progress and interpret the data in the results section thus it needs thorough revision as follows.
- It is necessary to summarize the experimental design in the beginning of this section:
Dear Reviewer - we present a summary - The presented results provide the basis for continuing research on the assessment of the impact of very low doses of ZEN on the macroorganism. There is a specific interaction between the mycotoxin level in the bone marrow microenvironment and the hematological values of the test animals. Preliminary data analysis documents that it is a set of highly individualized results for the parent substance and the absence of essential ZEN metabolites. The presented experimental design confirmed the correctness of the use of the method (chromatography) classified as precision medicine. Medicine that allows the assessment of a single organism as well as the entire population. The existing values of the concentration of the undesirable substance in the bone marrow microenvironment contributed to the negative values of the Pearson's r coefficient for white and red blood cells and platelets.
- In the section 2.1 it is written that “The analysed feed did not contain mycotoxins, or its mycotoxin content was below the sensitivity of the method”. Where are the data that prove this?
Dear Reviewer - please acknowledge the sentence presented because the provision presented in section 2.1. was entered as a result of the entry in section 4.2.1. where we presented the methodology of the conducted toxicological tests of fodder. As the results were below the sensitivity of the method (i.e., within the noise limits), it is difficult to present any meaningful results.
- The same concern is for the 2.2 section about “clinical observations”. If you refer to your previous studies only do not separate this observation in a separated section.
Dear Reviewer - The insertion of this section is a result of the Reviewers' comments in previous studies, where they wished to present the results of these observations in this form. On the other hand, the lack of such a statement requires us to present a detailed description of the method on which we determined the absence of clinical symptoms.
- What is presented in Table 2? The results of the control group? Why is it necessary to present the control results in a separate table? It would be more understandable to present the effect of ZEN administration only relative to the control samples. On the other hand the symbols used to present significance are not conventional. It would be better to use e.g. a,b,c letters instead.
Dear Reviewer - we think so too, but not everyone thinks so. In such a situation, we would suggest transferring the indicated Table 2 unchanged to Supplementary Tables.
- Regarding Figure 1 and 2 it would be enough to use the data of the tables only, the figure is not necessary. Regarding the statistics comparing the results of ZEN groups with the control group is the most relevant in order to see the effect of ZEN administration. In the present form of the table this relevance is hard to interpret. Please repeat the statistical analysis in the case of MCHC and HGB (in Figure 1) and in the case of RBC (in Figure 2).
Dear Reviewer - we would propose to leave both Figures just because the results are presented to Group C and you can see that the SD values are very small. This is probably why, however, statistical differences were found. We checked the statistical analysis and the results repeated.
- Regarding Table 3, 4 and 5 it would be more understandable to use figures instead of tables. Why not the same parameters are presented in these tables? If you want to compare the effect of ZEN administration in different doses the same parameters need to be presented.
Dear Reviewer - a very large amount of data has resulted in the need for a minimalistic presentation of it. This influenced the decision to present initially only the Table with the results of Gruop C (as reference groups), and in the subsequent ones only those results of the experimental groups where there were statistical differences.
- It is not necessary to present what correlation means in the section 2.6. However, it is required to present the experimental design and to explain which factors were chosen to see correlations. Only those that are presented in Table 7? Why these?
Dear Reviewer - all comments or suggestions have been included in the text with justification.
- The discussion section should provide an explanation for the observed results. In the present form it repeats the results and contain a lot of information that should be relocate to the results section including Figure 3 that would make the results more understandable. Only the explanation of the data should remain in this section.
Dear Reviewer - the text of the discussion has been redrafted as follows: (i) we have resigned from all numerical data, indications from Table or Figures, and from specifying statistical values; (ii) we shortened the whole text by nearly 30 lines; and (iii) we moved Figure 3 to the results section.
- What is the practical importance of your data? E.g. in breeding animals? It should be discussed in the conclusion section.
Dear Reviewer - we have slightly re-edited this section, removing numbers and other symbols from it. Due to the fact that we work at the Veterinary Faculty and the need for a practical reference, we presented this point of view in the last paragraph of section 3.5.

Reviewer 2 Report
- In the section materials and methods was completely absent a description of the methods used for the sample preparation (tissue extraction) and the quantitative analysis. In the section tissues samples, was only cited that the “extraction procedure was carried out in accordance with the recommendation…..”. (line It is not sufficient, please add in the text the detailed description of methods used. The same at section 4.3.1.3, the authors explain that the quantitative analysis of Zen were performed by LC-Ms. Which is the method used? The gradient elution? The type of detector used (Dad; fluorometric, IR)?
- The layout of Table 1 must be improved to facilitate the interpretation.
Author Response
Open Review
(x) I would not like to sign my review report
( ) I would like to sign my review report
English language and style
( ) Extensive editing of English language and style required
( ) Moderate English changes required
(x) English language and style are fine/minor spell check required
( ) I don't feel qualified to judge about the English language and style
|
Yes |
Can be improved |
Must be improved |
Not applicable |
|
|
Does the introduction provide sufficient background and include all relevant references? |
(x) |
( ) |
( ) |
( ) |
|
Is the research design appropriate? |
(x) |
( ) |
( ) |
( ) |
|
Are the methods adequately described? |
( ) |
( ) |
(x) |
( ) |
|
Are the results clearly presented? |
(x) |
( ) |
( ) |
( ) |
|
Are the conclusions supported by the results? |
(x) |
( ) |
( ) |
( ) |
Comments and Suggestions for Authors
- In the section materials and methods was completely absent a description of the methods used for the sample preparation (tissue extraction) and the quantitative analysis. In the section tissues samples, was only cited that the “extraction procedure was carried out in accordance with the recommendation…..”. (line It is not sufficient, please add in the text the detailed description of methods used. The same at section 4.3.1.3, the authors explain that the quantitative analysis of Zen were performed by LC-Ms. Which is the method used? The gradient elution? The type of detector used (Dad; fluorometric, IR)?
Dear Reviewer - all methodological suggestions have been made directly in the text in green.
- The layout of Table 1 must be improved to facilitate the interpretation.
Dear Reviewer, we have introduced changes to the description of Table 1 marked in green, which, in our opinion, should facilitate the reading and interpretation of the presented results.

Round 2
Reviewer 1 Report
The authors reaplied to all my concerns.